# Prospers and Obstacles in Using Artificial Intelligence in Saudi Arabia Higher Education Institutions—The Potential of AI-Based Learning Outcomes

**Nayef Shaie Alotaibi [1],\* and Awad Hajran Alshehri [2]**

1   College of Languages and Translation, Imam Mohammad Ibn Saud Islamic University, P.O. Box 5701, Riyadh 11432, Saudi Arabia
2   Department of English, College of Languages and Translation, Imam Mohammad Ibn Saud Islamic University, Riyadh 11432, Saudi Arabia
\*   Correspondence: nsotaibi@imamu.edu.sa

**Abstract:** Within the framework of the ongoing implementation of the 2030 Vision for Comprehensive Development of Higher Education in Saudi Arabia, the integration of artificial intelligence (AI) has emerged as a pivotal objective for the country's numerous higher education institutions. This study aims to examine the opportunities and challenges that arise from the adoption of AI-based learning outcomes in Saudi Arabia's higher education institutes. Moreover, the research also investigates the contribution of major higher education institutes in Saudi Arabia to the field of AI-based learning outcomes. To gather relevant literature, the Scopus and Web of Science databases were utilised, resulting in the selection of fifty-five studies for final analysis. The study employed the PRISMA statement 2020 for records filtration and utilised VOS viewer software to classify the literature on AI-based learning outcomes in Saudi Arabian universities. Through detailed analysis, three significant data streams were identified and examined. The findings indicate that AI is in a nascent stage within the realm of learning, and it has become an undeniable reality for higher education institutions. Embracing this transformative technology is crucial for meeting future learning challenges, and it is imperative that all students acquire the necessary technical skills to interact with and create artificial intelligence in the future. According to the findings, AI has the potential to address significant educational challenges, revolutionise teaching and learning methodologies, and accelerate progress toward the Saudi 2030 objectives. However, the study also highlights certain challenges associated with the implementation of AI-based learning in the higher education context of Saudi Arabia, emphasising the need for teachers to acquire new technological skills to effectively utilise AI pedagogically.

**Keywords:** artificial intelligence; learning outcomes; higher education institutes; PRISMA 2020; learning environment; Saudi Arabia

## 1. Introduction

Since the outbreak of the pandemic, schools and institutions have been obliged to move most of their instruction online [1]. On the surface, this has sparked a wave of educational innovation. Institutions worldwide have welcomed increased flexibility, providing virtual and actual courses [2]. However, before recent technological advancements, the learner-centred education system was a step from the old tutor-centred education system [3]. Education must evolve to meet changing lifestyles, the economy, technology, and student demands. Additionally, in recent years, with the advancement of educational technologies, the application of artificial intelligence (AI) in education has attracted significant interest from the public, governments, and academia [4]. According to Kuleto, Ilić, et al. [5], colleges and universities provide equal access to higher education by implementing AI and other technological advancements, disregarding geographical boundaries and time con-

straints. This enables students to engage in a learning environment that prioritises creativity, cooperation, and curiosity, developing global students and borderless universities.

The rapid progress of these technological innovations has undeniably exerted a profound influence on the domain of education, granting students novel capabilities and fostering a collaborative learning milieu within Higher Education Institutes (HEIs). Consequently, this has yielded significant implications that are poised to shape the foreseeable future [6]. In addition, most prominent higher education institutions have recognised that AI represents the present and future of education and global growth [7]. Such technologies give pupils an engaging and enhanced educational experience. According to Chatterjee and Bhattacharjee, [8], extra attention must be paid to a few key factors to achieve high academic quality. Researchers believe there is an urgent need to use cutting-edge technology, such as artificial intelligence. In addition, learning may be personalised with the aid of AI. It can meet the unique requirements of all student groups [9,10]. Every student would appreciate obtaining a wholly new and distinct educational method that is suited to the student's particular needs. AI-powered libraries can improve learning experiences at higher education institutions [11]. Accordingly, Hwang et al. [12] suggested that one of the most important goals of AI in education is to provide individual students with customised learning recommendations or aids depending on their learning status, preferences, or personal attributes. From the standpoint of precision education, which emphasises the importance of providing prevention and intervention practices to individual learners by analysing their learning status or behaviours, allowing learning systems to serve as intelligent tutors by incorporating experienced teachers' knowledge and intelligence into the system's decision-making process is a critical issue [13].

Integrating artificial intelligence in education has introduced novel opportunities for generating more effective learning experiences and improved technology-based learning environments or applications [14]. The other significant challenge in implementing AI-based learning is the infrastructural gap between developed and underdeveloped countries [15]. Meanwhile, some countries are adopting these technologies to increase the skills and capacities of their educational infrastructure to satisfy the requirements of higher education establishments [16]. According to Alhubaishy and Aljuhani, [17], Saudi Arabia is a developed country with several public and private colleges founded during the previous two decades. Saudi Arabia has been carrying out the National Transformation Program (NTP) as part of a national goal known as Saudi Vision 2030. The NTP's primary goal is to build and speed the execution of "digital infrastructure projects" in Saudi Arabia's higher education institutes. In addition, in educational institutions, Artificial Intelligence has been used to ease the learning process in terms of its ability to overcome various difficulties, such as time and capacity in conventional learning [18].

Furthermore, Saudi Arabia has prioritised national development objectives to make considerable progress in the research of AI [19]. The number of research articles published in international scientific publications and authorised patents must be used to gauge the progress of this procedure [20]. Universities have extensive worldwide scientific research capabilities and collaboration potential to extend research-based learning for the students and teachers at Saudi universities [21]. While there is growing interested in the use of artificial intelligence (AI) in education, there is a lack of comprehensive research focusing specifically on the opportunities and challenges associated with implementing AI-based learning outcomes in Saudi higher education institutes [22]. While Saudi Arabia has made substantial efforts towards national development goals and prioritised the use of AI in several areas, including education, the deployment of AI-based learning outcomes at Saudi Arabian institutions is still in its early phases. As a result, there is a research gap in understanding the unique potential and difficulties that exist in the Saudi Arabian context for incorporating AI-based learning outcomes in higher education. However, implementing AI-based outcomes in Saudi Arabia universities is still in its early stages. The objective of this study is to investigate the opportunities and challenges related to the implementation of AI-based learning outcomes in higher education institutes in Saudi Arabia. Additionally,

the study aims to examine the research contributions made by major higher education institutes in Saudi Arabia in the field of AI-based learning outcomes. The research will utilise data extraction from the Scopus and Web of Science databases, following the research methodology outlined in the PRISMA statement 2020.

## 2. Research Methodology

The PRISMA Statement, which provides guidelines for reporting systematic reviews and meta-analyses, has been in existence for over ten years. Over time, it has gained widespread recognition as one of the most notable reporting recommendations for social science research [23]. Since its original publication, the PRISMA Statement has undergone several extensions, which have been released in response to advancements in knowledge synthesis technologies [24]. This research has used the PRISMA Statement 2020 to include and exclude articles from the current investigation.

In December 2022, a comprehensive online search was conducted utilising the Scopus and Web of Science databases. These databases were chosen due to their extensive coverage of up-to-date articles, book chapters, and review papers. The search focused on the period from 2011 to 2021, with a particular emphasis on recent reports encompassing studies on the application of artificial intelligence in higher education institutions. To identify appropriate and relevant records for the study, a thorough analysis of the selected databases was performed. Multiple queries were utilised during this search, incorporating terms such as "Artificial intelligence", "Higher Education Institutions", "learning outcomes", and "Saudi Arabia". The selection criteria for this study were twofold: (a) the studies had to be specifically centred on the application of artificial intelligence in higher education, and (b) only studies published in the English language were considered.

Furthermore, two additional criteria were established to ensure the suitability of the selected papers: (c) the papers must have full-text availability, and (d) they must have been published after 2010. The initial selection of papers was based on the primary criteria of keywords "Artificial intelligence" and "Higher Education Institution", resulting in a total of 4038 articles. The second criterion of discipline narrowed down the selection to 3011 published articles. In the fourth stage of the PRISMA process, a further refinement of the selection was conducted, taking into consideration subjects directly relevant to the topic. Disciplines such as Computer Science, Mathematics, Engineering, Social Sciences, Materials Science, Environmental Science, Business, Management and Accounting, Psychology, and multidisciplinary studies were identified as significant contributors to the current study. Consequently, in the fifth stage of the selection process, criteria were established regarding the type of publication, with only review articles (6), book chapters (5), and research articles (53) included for analysis, resulting in a total of 104 pieces (as depicted in Figure 1). Finally, the fifth criterion was based on scientific papers solely from Saudi Arabia, leading to the analysis of 55 articles from both databases. It is important to note that during the selection process, careful attention was given to evaluate record duplication, excluding irrelevant materials, and including only those papers with complete document details. The selection and rejection criteria based on the PRISMA statement 2020 are presented in Figure 1.

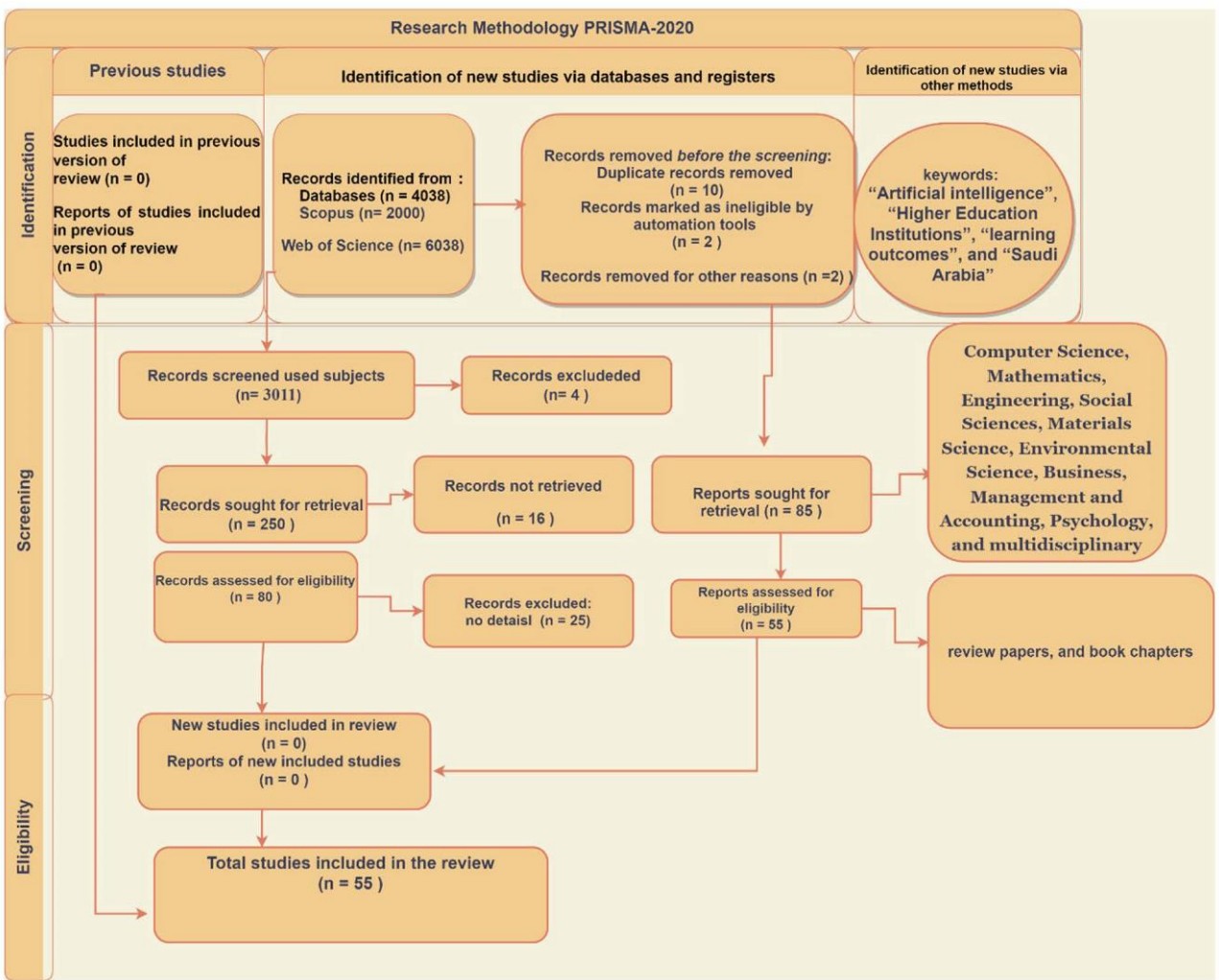

**Figure 1.** PRISMA statement 2020 inclusion and exclusion criteria.

## 3. Analysis

### 3.1. Descriptive

The subject criteria used for the extraction of records for the data analysis, computer sciences (31%), mathematics (20%), engineering (16%), and social sciences (13%) are significant subject areas that contributed to the current study. On the other hand, Figure 2 illustrates the number of low articles selected from multidisciplinary, business, management, accounting, environmental sciences, and psychology publications.

Figure 3 also depicts selected publications released after 2010. The majority of publications were published between 2019 and 2022, with only a few studies released between 2010 and 2018. This demonstrates that the number of publications increases year after year, demonstrating that higher education institutes in Saudi Arabia are becoming more interested in artificial intelligence operation and learning. Recent technology improvement has been a crucial catalyst for university AI-based learning results. Figure 3 depicts the number of papers chosen each year.

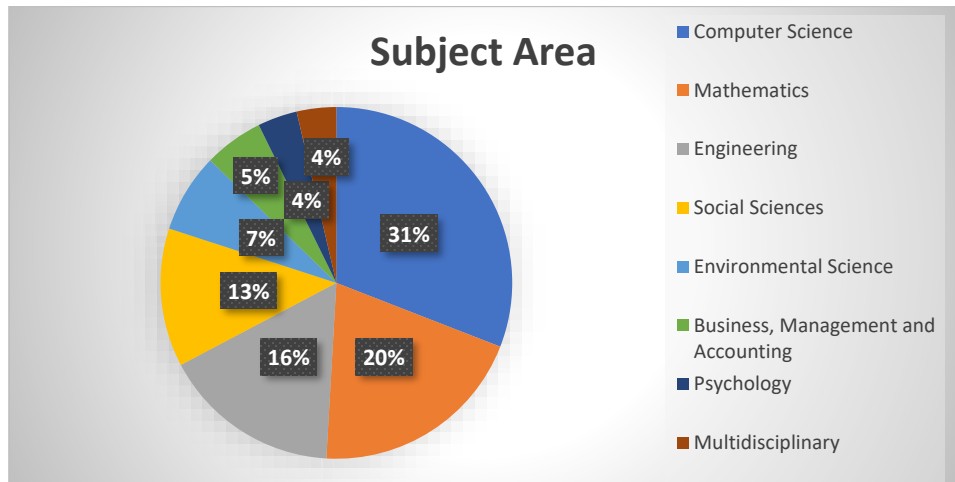

**Figure 2.** Distribution of documents from each subject.

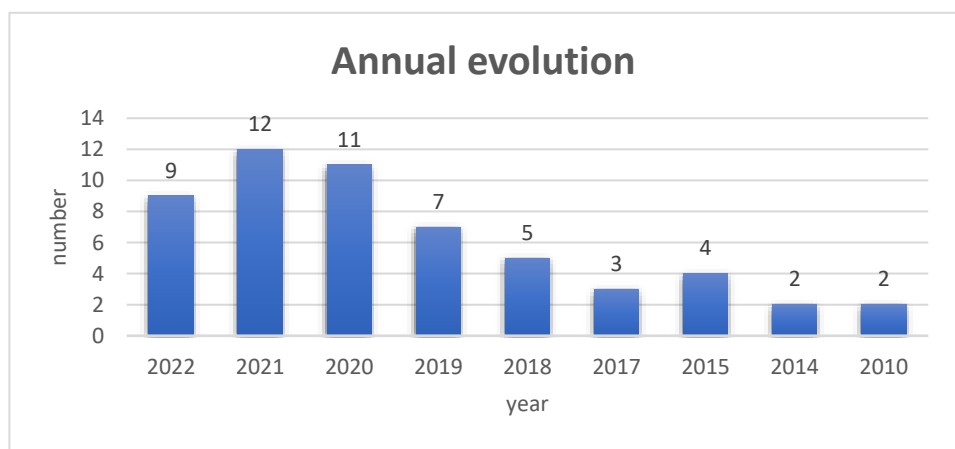

**Figure 3.** Distribution of documents from Annual evolution.

Furthermore, the present study's source-based publishing analysis indicates that IEEE Access provided the most citations, with an average of 60% of the total and seven articles, which is also the most from the source. The International Journal of Advanced Computer Science and Applications is second on the list of contributors, with five publications and 2% of all citations. The prior studies' names are gradually being deleted from the present study, and further key contributions are emphasised in Table 1 with the source's title, the number of publications chosen, the number of times mentioned, and the average number of citations each time.

In parallel, we examined how Saudi Arabia's higher education institutions contributed to the learning outcome of Artificial Intelligence. King Abdulaziz University's institutional contribution to the adaptation of AI-based learning outcomes research is considerable, with thirteen documents at the top of the list. Al Qasim University is a substantial contributor to this study, contributing seven records, as seen in Figure 4. King Khalid and Umm Al-Qura Universities also provided significant contributions in the shape of good publications. Despite these contributions, however, there is still a scarcity of high-quality research produced by Saudi Arabia's finest universities.

**Table 1.** Source titles, number of articles cited, and average citations.

| Source Title | Number of Articles | Cited by | Average Citations |
|---|---|---|---|
| Artificial Intelligence Review | 2 | 56 | 6% |
| Engineering Applications of Artificial Intelligence | 4 | 118 | 13% |
| Frontiers of Computer Science | 2 | 8 | 1% |
| IEEE ACCESS | 7 | 537 | 60% |
| Intelligent Automation and Soft Computing | 3 | 2 | 0% |
| International Journal of Advanced Computer Science and Applications | 5 | 14 | 2% |
| International Journal of Educational Technology in Higher Education | 1 | 99 | 11% |
| International Journal of Emerging Technologies in Learning | 2 | 26 | 3% |
| International Journal of Higher Education | 2 | 4 | 0% |
| Journal of Theoretical and Applied Information Technology | 3 | 6 | 1% |
| Sustainability (Switzerland) | 2 | 14 | 2% |
| World Wide Web-Internet and Web Information Systems | 2 | 14 | 2% |

*3.2. Content Analysis*

　　Artificial Intelligence is relatively new for Saudi Arabian universities, and researchers are looking deep into these changes step by step. In the current study evaluating artificial intelligence's effect on the learning process in Saudi Arabia's higher education institutes, we applied different keywords to extract the relevant material. Further categorisation of literature influences was conducted on the published literature and researcher perspectives on AI adaption in Saudi Arabian universities. We found the literary occurrences by finding the most often-used terms in the research. Table 2 displays a collection of artificial intelligence (AI) keywords and their categorisation, as well as the number of occurrences and relevance ratings. The value or significance of each phrase in the context of the research or study is indicated by the relevance score. For example, the phrase "new technology" appears 11 times in the dataset and has a high relevance score of 2.4656, indicating that it is a major and relevant term. The term "institution" on the other hand has 53 occurrences but a low relevance score of 0.2243, indicating that it may not be as central or essential to the research. The table also contains terminology associated with specific AI fields or ideas, such as "big data", "machine learning", and "Internet". These words have different occurrences and relevance ratings, which provide information about their frequency and importance in the dataset.

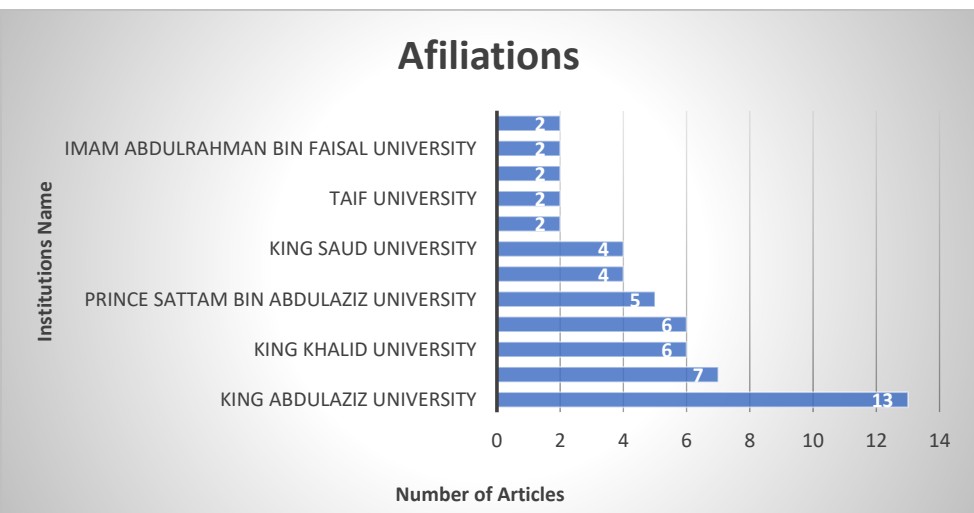

**Figure 4.** Distribution of documents from each institution in Saudi Arabia.

Furthermore, terminology linked to higher education, teaching, and learning is included in the table, such as "course", "teacher", and "ability". These words have varying occurrences and relevance ratings, indicating their importance and relevance in the context of AI in education. As previously stated, 55 studies were included in the keywords in the first stage of the literature review; these studies were then utilised to identify the literature categories from these keywords. The VOS viewer soft critical is used to categorise the occurrences of the most common words and the 48 critical phrases detected. Each term appears at least 10 times in the documents chosen. Furthermore, the 4 key themes were established based on term occurrences, and the relevance score of each phrase is highlighted in Table 2.

A more in-depth investigation of the records using content analysis was performed to establish and validate the research categories. The published literature was examined using VOS Viewer software, which groups the text's data into clusters based on related themes. According to a recent study, keywords used by researchers and those added later in the databases' indexation of journals are both relevant for bibliometric analyses to identify the structures of an investigation's field [1]. We used both keywords for the co-occurrence analysis within the research domain related to social entrepreneurship. The study included 55 documents, and the data contained 48 keywords. We carefully considered and selected only the 48 keywords that appeared at least 10 times. The results of the major classification themes are displayed in Figure 5. The group identified 4 significant clusters in Figure 5 in different colours. Research related to machine learning, algorithm, quality and strategy is shown in the yellow cluster. The blue hue represents expansion, factor, danger, change, and society. We also have grey themes for large data, new technology, higher education, and ability. Finally, the green collection stands for creativity, learning, and artificial intelligence. Figure 5 depicts the precise key-term occurrences and categorisation tailored for the current inquiry.

**Table 2.** Keyword occurrences, classification, and relevance score.

| Term | Classification | Occurrences | Relevance Score |
| --- | --- | --- | --- |
| area | | 26 | 0.7283 |
| big data | | 15 | 0.9354 |
| course | | 38 | 1.0278 |
| factor | | 20 | 0.8431 |
| future | | 24 | 0.9699 |
| Internet | | 16 | 0.9271 |
| need | | 34 | 0.807 |
| new technology | | 11 | 2.4656 |
| number | Artificial intelligence | 15 | 0.9013 |
| order | | 24 | 0.6057 |
| performance | | 28 | 1.3028 |
| resource | | 20 | 1.1105 |
| strategy | | 16 | 0.5043 |
| term | | 16 | 0.6555 |
| topic | | 18 | 1.0006 |
| trend | | 24 | 0.8182 |
| way | | 29 | 0.7726 |
| addition | | 12 | 1.3876 |
| difference | | 14 | 0.6537 |
| effect | | 19 | 0.5861 |
| implementation | | 26 | 0.6757 |
| information | | 20 | 0.8676 |
| institution | Higher education | 53 | 0.2243 |
| problem | | 31 | 0.5136 |
| quality | | 19 | 1.1374 |
| risk | | 15 | 1.3289 |
| teacher | | 25 | 1.3655 |
| teaching | | 37 | 0.598 |
| ability | | 18 | 1.855 |
| content | | 23 | 1.3973 |
| country | | 14 | 1.1383 |
| example | | 14 | 0.9882 |
| experience | | 22 | 0.9603 |
| issue | Learning | 27 | 0.5997 |
| learner | | 12 | 0.7668 |
| literature | | 19 | 0.7351 |
| relationship | | 19 | 0.4109 |
| skill | | 32 | 0.5786 |
| technique | | 29 | 1.4325 |

**Table 2.** *Cont.*

| Term | Classification | Occurrences | Relevance Score |
|---|---|---|---|
| accuracy | | 17 | 2.6668 |
| algorithm | | 30 | 1.6456 |
| concept | | 16 | 0.7479 |
| field | | 34 | 0.6087 |
| industry | Machine learning | 18 | 1.0749 |
| machine | | 18 | 1.8838 |
| machine learning | | 21 | 0.4868 |
| Opportunity | | 18 | 1.1174 |
| Year | | 24 | 0.7813 |

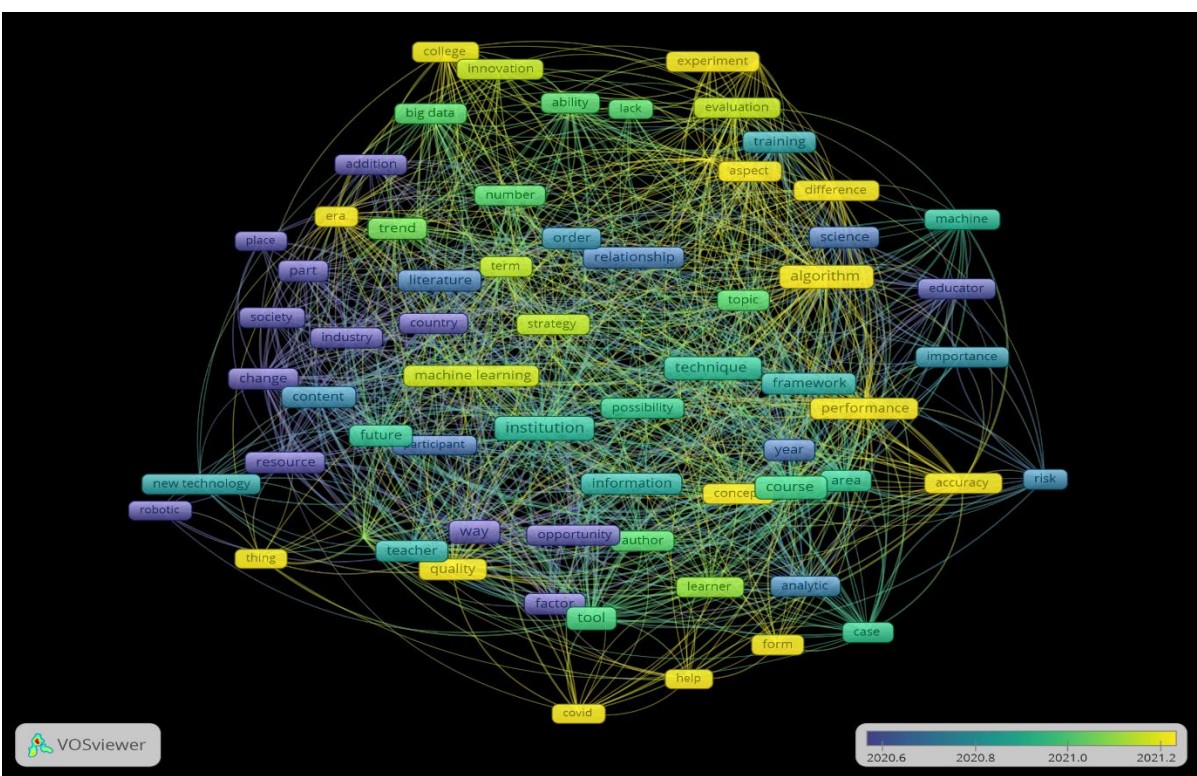

**Figure 5.** Distribution of data streams.

## 4. Classification

### 4.1. Higher Education

Saudi Crown Prince Mohammed Bin Salman launched an ambitious national plan, Saudi Vision 2030, in 2016, emphasising education as a critical component of the kingdom [25]. In addition, Saudi Arabia's higher education system is centralised. While it has not been expressly stated that Saudi Arabia's education system is centralised, various management and leading processes and practices have depicted such centralisation [26]. In comparison to other nations in the area, Saudi Arabian universities are adopting innovative digital technology at a rapid pace. The digital transformation phase drastically alters teachers' and students' learning processes and skill sets [27]. Conventional classroom instructors desperately require training to comprehend the contrasts between online and traditional education [28]. That new normal condition necessitates that all language teachers modify their teaching characteristics by finding and mastering new tactics and technical skills.

If the instructors need to learn how to incorporate technology into their teaching using appropriate instructional skills, their technological abilities are sufficient [29]. However, in today's world, a technological revolution is unavoidable for universities, governments, professors, and students [30]. According to Gürdür Broo et al. [31], higher education digital transformation is significantly influenced by government policies and institutional development plans. The process of digital transformation in higher education seeks to redefine educational offerings and restructure operational procedures. Table 3 below shows the details of authors, citations of articles, sub-classification details, segments and settings in which the research was conducted.

**Table 3.** Authors, cited by sub-classification, segments, and settings.

| Authors | Cited by | Sub-Classification | Segments | Settings |
|---|---|---|---|---|
| Khairy et al. [32] | 2 | learning environment | Educational Robotics' (ER) | Artificial intelligence |
| Nurunnabi et al. [25] | 18 | COVID-19 | health and well-being | university students |
| Al-Daraiseh et al., 2015 [27] | 17 | Artificial intelligence | optimise energy consumption | system and optimise |
| Mustapha et al. [30] | 17 | Industry 4.0 | disruptive advances | engineering education |
| Fayoumi and Hajjar, [33] | 11 | Artificial intelligence | academic performance | artificial neural network (ANN) model |
| Mohammad Abu-dalbouh, [28] | 2 | teaching and learning | (SARS-CoV-2) | distance learning |
| Assiri et al. [34] | 5 | Academic advising | students' academic problems | Artificial Intelligence |
| Aiyed and Aldosari, 2020 [35] | 4 | Artificial intelligence | applications in education | environment |
| Omar et al. [36] | 4 | COVID-19 | digital transformation | governance practices |
| Alshaikh et al. [26] | 1 | university's resources | collaborative filtering technique | recommender system (RS) |
| Hooda et al. [37] | 9 | COVID-19 | online education | learning and teaching |

The Kingdom of Saudi Arabia has recently implemented digital projects to modernise higher education institutes. These projects aim to enhance operations and the value of universities with the end objective of attaining sustainable development and worldwide effectiveness, increasing the digital economy's contribution to GDP, and improving the overall quality of the education system [36]. According to (Fayoumi and Hajjar, [33], incorporating new data mining and decision-making approaches in higher education is a bold step toward improved performance. Predictive and descriptive analytics provide intriguing insights into essential elements of education. According to studies by Hooda et al. [37], AI assists HEIs in increasing educational quality by boosting students' ultimate outcomes. Today's students will become tomorrow's leaders. Higher education executives who understand the benefits of AI must equip their institutions with AI that can assess students, offer feedback, and test scientific theories just as effectively as a person can [32]. In addition, AI is also helpful in academic advising in universities; Intelligent Academic Advising is a new artificial intelligence trend that seeks to automate academic advising responsibilities. According to Assiri et al. [34], Intelligent reporting systems generate individualised guidance and long-term educational planning by utilising algorithms, resource-intensive databases, and complicated queries.

Furthermore, artificial intelligence is still a relatively new issue in the Arab world. Arab universities follow a traditional approach to education, with the infrastructure of these universities varying. Hence, there is no applied research, studies, or even theory on the topics of artificial intelligence [38]. However, research efforts were limited to

shedding light on the availability of education technology in these universities, although the plans of these universities indicate the possibility of adopting artificial intelligence as one of the means that support education [35]. Artificial intelligence applications are essential in many fields. Notwithstanding, they are vital for educational institutions and universities, a great necessity that cannot be overlooked [39]. Universities today are no longer limited to education but have evolved into a critical part of the system of sustainable development in societies, as is emphasised by [40]. Finally, artificial intelligence benefits students and teachers since it is utilised to build an educational environment and promote collaborative learning. The use of artificial intelligence and current technologies may assist instructors and students to gain more educational experience, as well as giving information to teachers and management about the practices and scope of artificial intelligence in education necessary to achieve greatness.

### 4.2. Learning Outcome

Higher education institutions gather unprecedented data, ranging from logs acquired by the institutional virtual learning environment to library access frequency [41]. Behind these efforts is the notion that the analyses will better understand student learning progress, leading to interventions to improve teaching and learning [42]. In accordance with Mujtaba et al. [43], the concept of imperfection in evolutionary settings is referred to as Imperfect Evolutionary Systems (IES) from now on. IESs are dynamic settings that are unpredictable and chaotic (changes can occur quickly at any time) and present several obstacles to the learning of the persons who live in them. To address the above-mentioned issues, several recent studies have turned to poorly assisted supervised learning [23]. On the one hand, some individuals employ structured or semi-structured data from online encyclopaedias to train Named Entity Recognition (NER) models. Wikipedia, the world's most enormous open-world information base, has many poorly annotated texts with internal linkages of entities and a well-structured knowledge base of diverse areas [44]. In addition, the obligation to act based on learning analytics results and who is accountable for student achievement by working on learning analytics are critical concerns that must be carefully evaluated about student expectations [45]. Table 4 below shows the details of authors, citations of articles, sub-classification details, segments, and settings in which the research was conducted.

**Table 4.** Authors, cited by, sub-classification, segments, and settings.

| Authors | Cited by | Sub-Classification | Segments | Settings |
|---|---|---|---|---|
| Yamani et al., 2021 [46] | 0 | digital competency | digital learning | big data |
| Li et al., 2020 [44] | 0 | training and selection | large training data | online encyclopaedia |
| Tanveer et al. [21] | 9 | Artificial intelligence | teachers' skills | Education for sustainable development |
| Almufarreh et al. [47] | | e-learning | mobile devices and ubiquitous computing | Blackboard system |
| Whitelock-Wainwright et al. [41] | 4 | service implementations | learning analytics services | ethical and privacy elements |
| Sousa et al., 2021 [48] | 2 | learning analytics | educational levels | critical assessment |
| Muniasamy and Alasiry, 2020 [49] | 25 | e-learning | algorithms and automated delivery | deep learning |
| Mujtaba et al. [43] | 2 | new environment | Particle Swarm Optimisation system | Artificial Neural Network |
| Rehman and Saba, [50] | 56 | learning algorithms | neural networks | future research |
| Brika et al. [51] | 0 | e-learning | COVID-19 | e-learning in higher education |

Furthermore, learning analytics has been widely investigated and employed in higher education institutions, owing to the maturity level of these institutions' use of data analysis technologies. However, despite some encouraging findings, learning analytics is not as widely used in other educational settings [48]. On the other hand, because of the introduction of information and communication technology (ICT) in the modern day, a plethora of applications in Higher Education have evolved (HE) [49]. In that setting, eLearning was introduced as a means of meeting the new set of educational expectations. eLearning is described as learning management systems that combine computer-mediated communication software features with online course content delivery methods Rehman and Saba, [50]. Furthermore, it is frequently referred to as the ability to read. It will make mixing learning more convenient, and given the popularity of "conventional" e-learning sites, it must be done. When educational and technical assets come together, this will be more than just a personal problem [51].

In a humanitarian context, the informatisation of education poses a substantial danger to current higher education, serving as a test of its pedagogical ability. The use of digital, virtual, and network information systems, high-performance computers of the new decades, and the expansion of the Internet (particularly the "fast Internet") are all significant advancements in education [47]. According to Yamani et al. [46], it is important to meet the requirement of learning technologies, assess the information science specialist training needs in Saudi universities, adopt a strategic planning approach, and develop the abilities and skills of digital technologies at Saudi universities. In addition, Artificial Intelligence, Big data, cloud computing, the Internet of things (IoT), cybersecurity, robots, additive manufacturing (3D printing), augmented reality, horizontal and vertical integration, and simulation enable technologies for Industry 4.0 higher education [51]. More specifically, the contribution related to AI and machine learning is crucial for the learning outcome in Saudi Arabia. According to [52], there is now an urgent need to support AI technology in learning in ways that will help society adapt to AI-driven financial requirements and practices. Ensuring effective AI learning requires more than just resources; it also requires evaluating and assessing what works in training and delivering data in ways accessible to teachers and suitable for optimisation.

### 4.3. AI-Based Learning

AI has significantly enhanced economic development by adding USD215 billion yearly gross value (GVA) to the Saudi economy by 2035 [49]. So far, the fundamental notion behind the Kingdom's Vision 2030 national growth strategy is the rapidly empowered technology sector (in government, business, services, industry, healthcare, and education) [53]. In addition, as technology advances, human-like robots and intelligent systems are employed to solve many problems in everyday life [54]. The importance of AI in automating these processes is understandable. Using AI-based technologies with the Internet has resulted in several educational advancements for instructors and students. Practitioners and scholars attempt to provide a dependable and efficient technique for improving the learning system [53]. According to Housawi et al. [55], AI significantly impacts many aspects of life, from innovative device applications to manufacturing, transportation, health, and other disciplines. AI technology is evolving, and its top limit is unknown. The influence of AI on education is likewise growing. AI is a vast field encompassing "computer science, cybernetics, information theory, neurophysiology, psychology, philosophy, linguistics, and other disciplines" [56].

According to Hemachandran et al., 2022 [57], students can study at their own rate using artificial intelligence instructors since these systems can be calculated based on the student's demands. Students may learn independently because no person has a set schedule. This would mostly be a single-user system, so students would be more open to asking questions because there would be no other students in front of whom they would be embarrassed or human teachers from whom they would be afraid of embarrassment [58]. On the other hand, the teachers' perspective illustrates that, when viewed through the eyes

of a teacher, they are the ones who are out of work. They will be the ones who will lose their jobs [59]. When we notice that artificially intelligent systems' engagement necessitates a large amount of human data, what happens once the systems acquire the data? The artificial intelligence systems would be trained using data from existing teachers and then applied to perform the same function as human tutors [60]. Table 5 below shows the details of authors, citations of articles, sub-classification details, segments, and settings in which the research was conducted.

**Table 5.** Authors, cited by, sub-classification, segments, and settings.

| Authors | Cited by | Sub-Classification | Segments | Settings |
|---|---|---|---|---|
| Bond et al. [58] | 297 | Internet of Things (IoT) | statistical/architectural trends | ongoing 5G initiatives |
| Alyahyan and Düştegör, [61] | 99 | Artificial intelligence | students | student attributes |
| Haddar et al. [62] | 46 | uses problem | Multidimensional Knapsack Problem | state-of-the-art heuristic methods |
| Jumani et al. [60] | 15 | research | Particle Swarm Optimisation (PSO) algorithm | publications |
| Sabbagh et al. [53] | 11 | virtual reality (VR) | VR simulation technologies | appropriate visual and haptic |
| Housawi et al. [55] | 5 | vision of 2030 | postgraduate medical training | learning analytics |
| Elgibreen and Aksoy, [59] | 4 | machine learning (ML) | learning process | improve the performance |
| Omara et al. [63] | 4 | Metric learning | Vector Machine | biometric recognition |
| Liu et al., 2022 [54] | 3 | AI-based machine | Learning processes | education |
| Jabeur et al., 2022 [56] | 3 | high education standards | eclectic entrepreneurship theory | future research opportunities |
| Latif et al. [64] | 1 | Artificial Intelligence (AI) algorithms | Student retention | machine learning algorithms |
| Muniasamy and Alasiry, [49] | | vision 2030 | alternative sources | Eastern Province |
| Sabbagh et al. [53] | | English language teaching | higher education level | machine learning, neural network |
| Hemachandran et al. [57] | | higher education | student assessment | educational sector |

Interestingly, AI-based methodologies might be utilised over 5G-IoT networks to improve performance at the application, physical, and network levels by forecasting traffic patterns on the web, hence enabling the provisioning of AI-based user apps [64]. In addition, technologies based on artificial intelligence have the potential to be utilised to improve equity and deliver individualised learning [61]. The study by Haddar et al. [62] suggested that AI will supplement rather than replace human scientific contributions, resulting in a formidable mix of talents. In addition, institutions that fight change and cannot adapt and those that do not embrace AI will lose competitiveness over time. They will have to create redundancies to survive, but they will eventually become redundant themselves. Change is the only eternal constant. Accepting change is critical for all higher education institutions in Saudi Arabia [63]. Additionally, academic leaders and researchers must consider the potential that what students learn today may be used to assess the economy instead of the market. The result of AI-based learning is critical for improving students' skills and capacities.

## 5. Results and Discussion

This study aims to explore the opportunities and challenges of implementing AI-based learning outcomes in higher education institutions in Saudi Arabia. In addition, it examines the research contributions of major higher education institutes in the context of AI-based learning outcomes in Saudi Arabia. To ensure a rigorous and systematic review process, this study employed the PRISMA statement 2020 for record inclusion and exclusion. Scopus and Web of Science databases were used to extract and select relevant records following a thorough filtration process. The study analysed the subjects, type of articles, year-based findings of records, and source-based citation averages in the second phase of the study. The primary aim of this study was to examine the involvement of Saudi Arabian universities in advancing AI-based learning outcomes in higher education. The results showed that King Abdulaziz University and Al Qasim University had made significant contributions to this field. Despite being in the early stages, there is a growing interest in adopting advanced learning methods. In the third phase of the study, VOS viewer software was employed to identify significant keyword occurrences and data streams, which were classified into three essential literature categories. These key term occurrences were then subjected to content analysis to validate their relevance.

The results indicate that higher education, learning outcomes and Artificial Intelligence-based learning are attracting the attention of the significant number of researchers discussed in the previous literature. Saudi Arabian universities are adopting these advanced technologies very quickly to improve the quality of learning in higher education institutes [65]. In addition, the results indicate that the Kingdom of Saudi Arabia has recently initiated programmes to modernise higher education establishments digitally. The results of Khairy et al. [32] suggested that AI is also helpful in university advising; Intelligent Academic Advising is a new artificial intelligence trend that seeks to automate academic advising responsibilities. In addition, according to Robles Carrillo, [40], universities are no longer restricted to teaching. They have also grown into an essential element of the system of sustainable development in communities, as this research emphasises. On the other hand, results indicate that learning analytics has been widely investigated and employed in higher education institutions, owing to the maturity level of these institutions' use of data analysis technologies. Similarly, because of the introduction of information and communication technology (ICT) in the modern day, many AI-based applications in higher education have evolved to enhance the learning outcome [66].

The results of Yamani et al. [46] suggested satisfying learning technology requirements, analysing information science expert training needs in Saudi universities, using a strategic planning method, and building digital technology abilities and skills in Saudi universities. Also, more importantly, the findings of Hemachandran et al. [57], suggested that since these systems may be computed based on the student's requests, students can learn at their own pace with artificial intelligence teachers. Because no one has a defined timetable, students may learn at their own speed. In contrast, when viewed through the eyes of a teacher, they are the ones who are out of work. They will be the ones who will lose their jobs [59]. However, Haddar et al. [62] concluded that AI would supplement rather than replace human scientific contributions, resulting in a formidable mix of talents.

## 6. Conclusions

The findings of this study provide significant contributions to the field of AI-based learning outcomes in higher education, particularly in the context of Saudi Arabia. The study reveals that Saudi universities are actively embracing artificial intelligence in the learning environment as part of their efforts to align with the Saudi Vision 2030. The results highlight the importance of integrating AI into higher education institutions to meet future learning challenges and enhance educational quality.

One significant contribution is the recognition that AI is an inevitable reality for higher education, and institutions must adapt and transform their approaches to incorporate AI technologies. The study emphasises the importance of equipping students with technical

skills to effectively engage with and contribute to the development of artificial intelligence. Furthermore, it suggests that primary education should focus on fostering skills where AI is less likely to have a competitive advantage, such as complex decision-making, critical thinking, entrepreneurship, and emotional intelligence.

The research also highlights the increasing popularity of AI applications and student assistance in higher education institutions. Machine learning is being utilised to guide students, automate course scheduling, and provide recommendations for courses, majors, and career options through guidance counselling or career services departments. This demonstrates the potential of AI to address pressing educational challenges, revolutionise teaching and learning methodologies, and accelerate progress towards the Saudi Vision 2030. Figure 6 below depicts the learning outcomes of the current study.

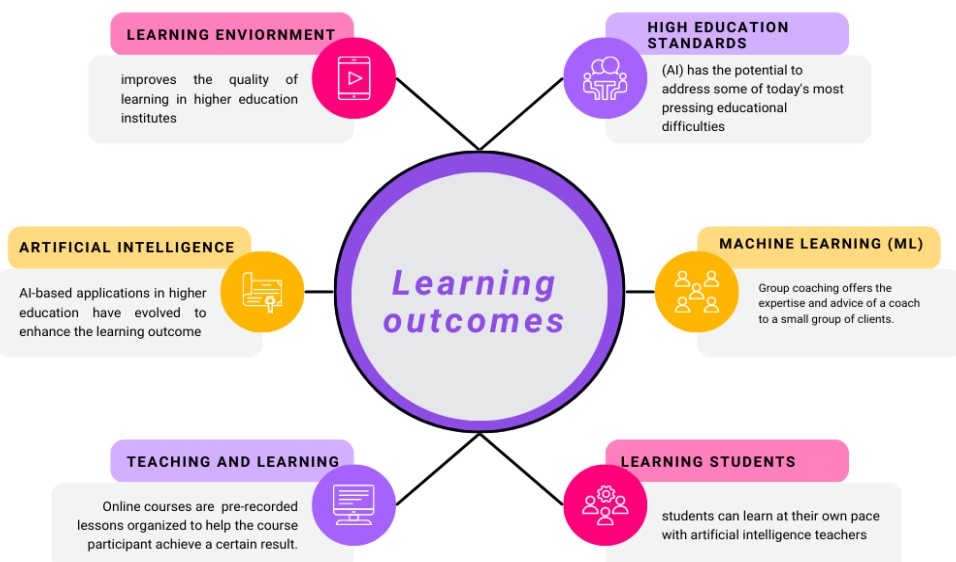

**Figure 6.** Outcomes of current study and future directions.

However, the study also acknowledges the challenges associated with implementing AI-based learning in the Saudi Arabian higher education context. Teachers need to acquire new technological skills to effectively utilise AI pedagogically. It is essential for instructors to master the tools and understand their applications when adopting AI systems at universities. Collaboration between AI developers and teachers is necessary to create sustainable solutions that align with real-world conditions. Additionally, addressing fundamental technical infrastructure gaps and allocating sufficient budgetary resources for software, hardware, and ongoing training are crucial factors for successful AI deployment in higher education.

## 7. Practical Implications and Future Agenda

Practical implications arising from this research include recommendations for higher education institutions, policymakers, and AI-based learning application developers. The study provides insights and guidance for Saudi higher education ministries to improve learning environments by leveraging AI technologies effectively. Furthermore, it emphasises the importance of preparing systems and data in advance to facilitate integration and further development of AI-based systems in education. Collaborations between researchers and the higher education industry are also encouraged to assist institutions in planning and implementing AI solutions efficiently, effectively, and ethically.

Overall, this research makes a significant contribution to the understanding of AI-based learning outcomes in higher education in Saudi Arabia, offering insights into the opportunities, challenges, and implications associated with the adoption of AI technologies in the learning environment.

**Author Contributions:** Conceptualization; methodology; software; validation, by N.S.A. and A.H.A.; formal analysis; investigation; resources; data curation, completed. In addition, writing—original draft preparation; writing—review and editing; visualization; supervision, was done by N.S.A. All authors have read and agreed to the published version of the manuscript.

**Funding:** This work was supported and funded by the Deanship of Scientific Research at Imam Mohammad Ibn Saud Islamic University (IMSIU) (grant number IMSIU-RG23087).

**Institutional Review Board Statement:** Not Applicable.

**Informed Consent Statement:** Not applicable.

**Acknowledgments:** Authors thank the support by the Deanship of Scientific Research at Imam Mohammad Ibn Saud Islamic University (IMSIU) (grant number IMSIU-RG23087).

**Conflicts of Interest:** The authors declare no conflict of interest.

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
