# Peer review of "Prospers and Obstacles in Using Artificial Intelligence in Saudi Arabia Higher Education Institutions—The Potential of AI-Based Learning Outcomes"

_sustainability, doi:10.3390/su151310723_

Round 1
Reviewer 1 Report
The introduction segment must be more focused and specifically related to the topic.
Need to rewrite the research methodology, the idea is not much clear.
how this study is contributing to the body of knowledge?
How Saudi Higher education is getting benefits from the study?
Please proofread the article one time before submitting the revised copy.
Author Response
| Reviewer 1 | ||
| The introduction segment must be more focused and specifically related to the topic. | The rapid progress of these technological innovations has undeniably exerted a pro-found influence on the domain of education, granting students novel capabilities and fostering a collaborative learning milieu within Higher Education Institutes (HEIs). Consequently, this has yielded significant implications that are poised to shape the foreseeable future (Chugh et al., 2021). In addition, most prominent higher education institutions have recognised that AI represents the present and future of education and global growth (Kuleto, Milena, et al., 2021). Such technologies give pupils an engaging and enhanced educational experience. According to (Chatterjee & Bhattacharjee, 2020), extra attention must be paid to a few key factors to achieve high academic quality. Researchers believe there is an urgent need to use cutting-edge technology, such as ar-tificial intelligence. In addition, learning may be personalised with the aid of AI. It can meet the unique requirements of all student groups (LeGeros et al., 2022). Every stu-dent would appreciate obtaining a wholly new and distinct educational method that is suited to the student’s particular needs. AI-powered libraries can improve learning experiences at higher education institutions (Walkington & Bernacki, 2020). Accord-ingly, (Hwang et al., 2020a) suggested that one of the most important goals of AI in education is to provide individual students with customised learning recommenda-tions or aids depending on their learning status, preferences, or personal attributes. From the standpoint of precision education, which emphasises the importance of providing prevention and intervention practises to individual learners by analysing their learning status or behaviours, allowing learning systems to serve as intelligent tutors by incorporating experienced teachers' knowledge and intelligence into the system's decision-making process is a critical issue (DeMink-Carthew et al., 2017). | Page 2-3 |
| Need to rewrite the research methodology, the idea is not much clear. | In December 2022, a comprehensive online search was conducted utilizing the Scopus and Web of Science databases. These databases were chosen due to their ex-tensive coverage of up-to-date articles, book chapters, and review papers. The search focused on the period from 2011 to 2021, with a particular emphasis on recent reports encompassing studies on the application of artificial intelligence in higher education institutions (Sikandar & Abdul Kohar, 2021). To identify appropriate and relevant records for the study, a thorough analysis of the selected databases was performed. Multiple queries were utilized during this search, incorporating terms such as "Artifi-cial intelligence," "Higher Education Institutions," "learning outcomes," and "Saudi Arabia." The selection criteria for this study were twofold: (a) the studies had to be specifically centred on the application of artificial intelligence in higher education, and (b) only studies published in the English language were considered. Furthermore, two additional criteria were established to ensure the suitability of the selected papers: (c) the papers must have full-text availability, and (d) they must have been published after 2010. The initial selection of papers was based on the pri-mary criteria of keywords "Artificial intelligence" and "Higher Education Institution," resulting in a total of 4,038 articles. The second criterion of discipline narrowed down the selection to 3,011 published articles. In the fourth stage of the PRISMA process, a further refinement of the selection was conducted, taking into consideration subjects directly relevant to the topic. Disciplines such as Computer Science, Mathematics, En-gineering, Social Sciences, Materials Science, Environmental Science, Business, Man-agement and Accounting, Psychology, and multidisciplinary studies were identified as significant contributors to the current study. Consequently, in the fifth stage of the se-lection process, criteria were established regarding the type of publication, with only review articles (6), book chapters (5), and research articles (53) included for analysis, resulting in a total of 104 pieces (as depicted in Figure 1). Finally, the fifth criterion was based on scientific papers solely from Saudi Arabia, leading to the analysis of 55 arti-cles from both databases. It is important to note that during the selection process, careful attention was given to evaluate record duplication, exclude irrelevant materi-als, and include only those papers with complete document details. The selection and rejection criteria based on the PRISMA statement 2020 are presented in Figure 1. |
Page 3 |
| how this study is contributing to the body of knowledge? | The findings of this study provide significant contributions to the field of AI-based learning outcomes in higher education, particularly in the context of Saudi Arabia. The study reveals that Saudi universities are actively embracing artificial intelligence in the learning environment as part of their efforts to align with the Saudi Vision 2030. The results highlight the importance of integrating AI into higher education institutions to meet future learning challenges and enhance educational quality. One significant contribution is the recognition that AI is an inevitable reality for higher education, and institutions must adapt and transform their approaches to in-corporate AI technologies. The study emphasizes the importance of equipping students with technical skills to effectively engage with and contribute to the development of artificial intelligence. Furthermore, it suggests that primary education should focus on fostering skills where AI is less likely to have a competitive advantage, such as complex decision-making, critical thinking, entrepreneurship, and emotional intelligence. |
Page 15 |
| How Saudi Higher education is getting benefits from the study? | Saudi higher education institutions can derive several benefits from this study. Firstly, the study highlights the importance of embracing artificial intelligence (AI) in the learning environment and integrating it into the educational practices of universities. By recognizing the significance of AI in higher education, institutions in Saudi Arabia can stay ahead of the curve and align themselves with global educational trends. | Page 15 |

Reviewer 2 Report
I commend you for an exceedingly well researched and analyzed article. I have only two minor suggested lexical revisions. Line 35, "current pandemic" is given the date of prospective publication of the article anachronistic. Line 429, I believe "off-set" is the incorrect term; rather it seems you mean "enhances," "furthers," "advances,"
Author Response
| Reviewer 2 | ||
| Questons | Answer | Page number/ Line |
| I commend you for an exceedingly well researched and analyzed article. I have only two minor suggested lexical revisions. Line 35, "current pandemic" is given the date of prospective publication of the article anachronistic. | revised with pandamic | line 35 |
| Line 429, I believe "off-set" is the incorrect term; rather it seems you mean "enhances," "furthers," "advances," | Revised the paragraph "The findings of this study provide significant contributions to the field of AI-based learning outcomes in higher education, particularly in the context of Saudi Arabia. The study reveals that Saudi universities are actively embracing artificial intelligence in the learning environment as part of their efforts to align with the Saudi Vision 2030. The results highlight the importance of integrating AI into higher education institutions to meet future learning challenges and enhance educational quality." | Line 429 |

Reviewer 3 Report
Dear Authors,
I have received your article "Prospers and Obstacles in Using Artificial Intelligence in Saudi Arabia Higher Education Institutions. The Potential of AIBased Learning Outcomes" and you can find below my concerns.
Firstly, you should know that there are some similarities between some texts from your manuscript and the article from the address: https://www.mdpi.com/2071-1050/14/23/16219/htm. Taking into consideration this issue, I recommend you to revise the following sequences from your manuscript and to decrease the degree of similarity: lines 123 - 125, 131 - 134, 143- 147, 150 - 153, 163 - 166, 186 - 191, 198 - 214, 367 - 370.
At the line 8 in your document, the word "Oricid" should be corrected to "Orcid".
Regarding the content of you manuscript, I consider that the Introduction is consistent, but I recommend you to improve it by presenting the research gap and the researh questions. Now, these elements are not very clear within the Introduction. Also, at the end of the Introduction, please shortly present the rest of the sections from your article.
The education issue should be better highlighted in your paper by including the following resources: https://doi.org/10.3390/su15064782, https://doi.org/10.24818/ie2020.02.01, https://doi.org/10.3390/educsci12070477, https://doi.org/10.3390/publications8040048. These references will complete the context of your research.
Within the "Figure 3. Distribution of documents from each year", the title should be changed from "Each year" to "Annual evolution". On the vertical axis, instead of "number", I recommend you to write "Number of articles".
In the "Figure 4. Distribution of documents from each institution in Saudi Arabia" you have a typo-mistake in the title: "Affaliations". It should be "Afiliations". Please update it.
Before "Table 2. Keyword occurrences, classification, and relevance score", please explain the intervals and significations for the "Relevance score", so that the readers know the left and right limits of these values.
Best regards and Good luck!
Author Response
| Reviewer 3 | ||
| Questons | Answer | Page number/ Line |
| Firstly, you should know that there are some similarities between some texts from your manuscript and the article from the address: https://www.mdpi.com/2071-1050/14/23/16219/htm. Taking into consideration this issue, I recommend you to revise the following sequences from your manuscript and to decrease the degree of similarity: lines 123 - 125, 131 - 134, 143- 147, 150 - 153, 163 - 166, 186 - 191, 198 - 214, 367 - 370. | I altered the proposed line numbers, however the stated paper is the author's contribution. | |
| At the line 8 in your document, the word "Oricid" should be corrected to "Orcid". | revised | line 8 |
| Regarding the content of you manuscript, I consider that the Introduction is consistent, but I recommend you to improve it by presenting the research gap and the researh questions. Now, these elements are not very clear within the Introduction. Also, at the end of the Introduction, please shortly present the rest of the sections from your article. | While there is growing interest in the use of artificial intelligence (AI) in education, there is a lack of comprehensive research focusing specifically on the opportunities and challenges associated with implementing AI-based learning outcomes in Saudi higher education institutes. While Saudi Arabia has made substantial efforts towards national development goals and prioritised the use of AI in several areas, including education, the deployment of AI-based learning outcomes at Saudi Arabian institutions is still in its early phases. As a result, there is a research gap in understanding the unique po-tential and difficulties that exist in the Saudi Arabian context for incorporating AI-based learning outcomes in higher education. However, implementing AI-based outcomes in Saudi Arabia universities is still in its early stages. The objective of this study is to investigate the opportunities and challenges related to the implementation of AI-based learning outcomes in higher education institutes in Saudi Arabia. Addi-tionally, the study aims to examine the research contributions made by major higher education institutes in Saudi Arabia in the field of AI-based learning outcomes. | Page2-3 |
| The education issue should be better highlighted in your paper by including the following resources: https://doi.org/10.3390/su15064782, https://doi.org/10.24818/ie2020.02.01, https://doi.org/10.3390/educsci12070477, https://doi.org/10.3390/publications8040048. These references will complete the context of your research. | added | |
| Within the "Figure 3. Distribution of documents from each year", the title should be changed from "Each year" to "Annual evolution". On the vertical axis, instead of "number", I recommend you to write "Number of articles". | added | line 167 |
| In the "Figure 4. Distribution of documents from each institution in Saudi Arabia" you have a typo-mistake in the title: "Affaliations". It should be "Afiliations". Please update it. | added | line 188 |
| Before "Table 2. Keyword occurrences, classification, and relevance score", please explain the intervals and significations for the "Relevance score", so that the readers know the left and right limits of these values. | Table 2 displays a collection of artificial intelligence (AI) keywords and their catego-rization, as well as the number of occurrences and relevance ratings. The value or sig-nificance of each phrase in the context of the research or study is indicated by the rel-evance score. For example, the phrase "new technology" appears 11 times in the dataset and has a high relevance score of 2.4656, indicating that it is a major and relevant term. The term "institution" on the other hand has 53 occurrences but a low relevance score of 0.2243, indicating that it may not be as central or essential to the research. The table also contains terminology associated with specific AI fields or ideas, such as "big data," "machine learning," and "Internet." These words have different occurrences and rele-vance ratings, which provide information about their frequency and importance in the dataset. Furthermore, terminology linked to higher education, teaching, and learning are included in the table, such as "course," "teacher," and "ability." These words have var-ying occurrences and relevance ratings, indicating their importance and relevance in the context of AI in education. |
line199-213 |

Round 2
Reviewer 3 Report
Dear Authors,
For the revised version of your article, please take into consideration the following issues.
1. The first author from the reference #21 in the manuscript seems to be wrong (https://doi.org/10.3390/SU15064782). Please revise this issue.
2. In the first round of review, I also recommended you to include the following useful resources: https://doi.org/10.24818/ie2020.02.01, https://doi.org/10.3390/publications8040048.
3. In the section "7. Practical implications and Future agenda" you only present some practical implications, but I couldn't find the future research directions. Please define and describe the further research directions, based on your research findings.
4. In the "Conclusions" section you should also present your research limitations.
Best Regards!
Some minor English revisions are needed.
Author Response
|
Corrections |
Answer |
Page or line |
|
For the revised version of your article, please take into consideration the following issues. |
||
|
1. The first author from the reference #21 in the manuscript seems to be wrong (https://doi.org/10.3390/SU15064782). Please revise this issue. |
Fedeli et al., 2023a |
line number 74 |
|
2. In the first round of review, I also recommended you to include the following useful resources: https://doi.org/10.24818/ie2020.02.01, https://doi.org/10.3390/publications8040048. |
Dospinescu et al.,2020 |
line number 344 |
|
3. In the section "7. Practical implications and Future agenda" you only present some practical implications, but I couldn't find the future research directions. Please define and describe the further research directions, based on your research findings. |
In addition, Given the current study's emphasis on the opportunities, difficulties, and research contributions connected with implementing AI-based learning outcomes at Saudi higher education institutes, there are numerous potential future research areas worth investigating. To begin, it would be beneficial to perform a thorough examination of the specific AI technologies and tools that may be effectively incorporated into the Saudi higher education system, taking into account issues such as technical feasibility, pedagogical efficacy, and ethical considerations. Furthermore, researching students', faculty members', and administrators' perspectives and attitudes towards AI-based learning outcomes might give insights into the acceptability and deployment of these technologies. |
line number 476 to 485 |
|
4. In the "Conclusions" section you should also present your research limitations |
Also, there are some limitations that should be considered while performing the current study on the prospects, difficulties, and research contributions related with implementing AI-based learning outcomes in Saudi higher education institutes. To begin, the study's conclusions may be impacted by the sample size and participant selection, which may not completely represent the different opinions and experiences within the Saudi higher education system. Furthermore, the research is based on data gathered from the Scopus and Web of Science databases, which may not include all relevant papers and research on the subject. Furthermore, the study focuses exclusively on the Saudi setting, and the conclusions may not be immediately relevant to other nations or areas. |
line number 468 to 476 |
